# Stochastic activations

## Abstract

We introduce stochastic activations. This novel strategy randomly selects between several non-linear functions in the feed-forward layer of a large language model. In particular, we choose between `SILU` or `RELU` depending on a Bernoulli draw. Our strategy circumvents the optimization problem associated with `RELU`, namely, the constant shape for negative inputs that prevents the gradient flow. We leverage this strategy in two ways:

(1) We use stochastic activations during pre-training and fine-tune the model with `RELU`, which is used at inference time to provide sparse latent vectors. This reduces the inference FLOPs and translates into a significant speedup in the CPU. Interestingly, this leads to much better results than training from scratch with the `RELU` activation function.

(2) We evaluate stochastic activations for generation. This strategy performs reasonably well: it is only slightly inferior to the best deterministic non-linearity, namely, `SILU` combined with temperature sampling. This offers an alternative to existing strategies by providing a controlled way to increase the diversity of the generated text.

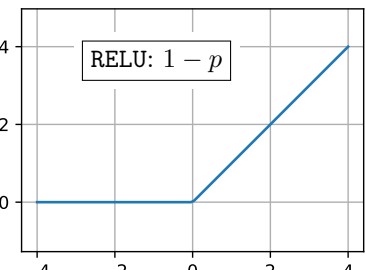 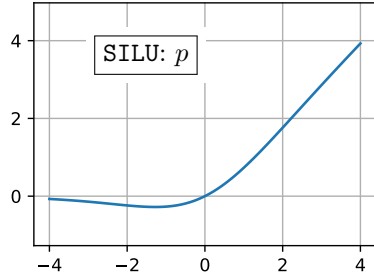

Figure 1: Stochastic activation randomly selects one of two activations when $x < 0$:
(1) `RELU` selected with probability $1 - p$; otherwise (2) another activation, in particular `SILU`.

## 1 Introduction

Large language models (LLMs) (Devlin et al., 2019; Chowdhery et al., 2022; Brown et al., 2020; Vaswani et al., 2017) have revolutionized natural language processing, enabling unprecedented capabilities in text generation, comprehension, and reasoning. Their success stems from scaling model parameters and leveraging vast amounts of data, but this comes with a significant computational complexity. As the demand for more efficient and powerful models grows, researchers are increasingly focused on optimizing their training processes to balance performance with resource constraints.

The majority of the LLM parameters are in the Feed-Forward Network (FFN) layers, where they primarily serve to store and recall information from the training data. FFNs are two linear layers separated by an *activation function*, and sometimes an additional linear layer that serves as a gating operation. The activation is a non-linear function from $\mathbb{R}$ to $\mathbb{R}$. In this context, the choice of activation function plays a crucial role for both the model's expressivity and efficiency. The simplest choice of activation is `RELU` (Rectified Linear Unit), that allows positive values to pass through and forces negative inputs to zero. `RELU` is *sparsity-inducing*,

since, on average, half of its outputs are zero (in practice significantly more). Within a two-layer Multilayer Perceptron, this means that inference on the second layer is a matrix-sparse vector multiplication, hence, it can be implemented with fewer FLOPs than a matrix-dense vector multiplication. Note that effectively improving the runtime with this sparsity pattern remains challenging.

In practice, the Sigmoid Linear Unit (`SILU`) activation, combined with a gated design, has consistently outperformed `RELU` in terms of model accuracy (Shazeer, 2020). Unfortunately, `SILU` does not induce sparsity. One plausible explanation for `RELU`'s underperformance is that its gradient for negative input values is zero, which hinders optimization by preventing weight updates in a significant portion of the network. Solutions like Leaky `RELU` (Maas et al., 2013) circumvent this problem by ensuring non-zero gradient almost everywhere, but they are inferior to `SILU` and involve abandoning sparsity. In contrast, if the so-called "dying `RELU` problem" optimization challenge could be effectively addressed, `RELU`'s theoretical advantages – such as sparsity and computational efficiency – may translate into performance comparable to `SILU` for a lower number of FLOPs. This disparity presents a challenge: how to harness the efficiency benefits of sparse activations, such as `RELU`, without sacrificing the empirical advantages of `SILU`. This motivates our exploration of alternative training strategies that mitigate `RELU`'s limitations while preserving its benefits.

In this work, we consider two ways to approach this problem. The first approach is activation fine-tuning, denoted by `Swi+FT`: we pre-train the model with an activation that facilitates efficient large language model optimization, then, we change the activation to `RELU` and adapt the model by fine-tuning it further. Our second approach, referred to as `StochA` (stochastic activations), is a novel technique that randomly selects between multiple activations, either at train or test time. Both approaches allow models to benefit from the superior optimization properties of `SILU`. These hybrid strategies combine the best of both worlds – maintaining high model performance while unlocking the computational efficiency of sparse activations.

In summary, this paper makes the following contributions:

- We introduce two strategies that employ different activation functions at training and inference time, namely `Swi+FT` and `StochA`. Both are complementary and make it possible to use activations at inference time that differ from those employed during pre-training.
- We produce `RELU`-based models that are much better than those obtained with regular training, i.e., our methods significantly outperform training with `RELU` only.
- We show that stochastic activations, when used at inference time, provide an alternative way to generate diverse sequences compared to traditional temperature sampling or other variants.

## 2  RELATED WORK

**Standard activation functions**  Activations are at the core of deep learning, they are fundamental to enable deep learning models to go beyond linear function with limited expressivity. While early neural network architectures were inspired by logistic regression, such as sigmoidal and tanh activations, many activation functions have been evaluated for the FFN layers of transformers. Vaswani et al. (2017) used `RELU` (Glorot et al., 2011) initially. However, using the `RELU` activation function leads to some neurons getting stuck in the negative region. As a consequence, models stop learning entirely, since the gradient is zero for negative inputs, and their weights do not get updated. In contrast, Touvron et al. (2023) used `SILU` as the activation function for the FFN layers of the transformer for the first Llama models. Shazeer (2020) discusses the benefits of `SWIGLU`, which consists of `SILU` with gating. There exist many other activation functions such as the Gaussian Error Linear Unit (`GELU`) (Hendrycks & Gimpel, 2016), Scaled Exponential Linear Unit (`SELU`) (Klambauer et al., 2017), `Swish` (Ramachandran et al., 2018) and gated `SILU`, among others.

In particular, the Leaky `RELU` (Maas et al., 2013; Redmon et al., 2015; Ridnik et al., 2021; Guo et al., 2024) tried tackling the dying `RELU` problem by allowing a small, non-zero

gradient when the input is negative in order to keep the neurons active and the gradients flowing, reducing the risk of dead neurons.

**Adaptive activation functions**  Lee et al. (2022) propose the Adaptive Swish(`ASH`) activation, which uses stochastic sampling of the top-k percentile elements. It generalizes the `Swish` (Ramachandran et al., 2018), which uses adaptive thresholding to select the values in the top percentiles and is set to zero otherwise. It is an example of a stochastic activation.

**Dropout and structured Dropout variants**  In the original dropout paper (Srivastava et al., 2014), the authors propose a regularization technique to reduce overfitting and improve generalization of a neural network. It consists of setting to zero a subset of neurons at each training step. Consequently, the dropped neurons do not contribute to the forward pass or receive weight updates during back-propagation. At inference time, all neurons are used and their outputs are scaled by the dropout probability.

LayerDrop (Fan et al., 2019) randomly drops entire layers during training, hence, it encourages the model to be robust to missing layers. At inference time, some layers can be pruned, trading off speed and accuracy as needed. While the method does not make the model sparse in the usual sense, it induces structured sparsity in the computation graph during training. Other works also introduce structured dropout variants such as DropBlock (Ghiasi et al., 2018), Bayesian dropout (Gal & Ghahramani, 2016), or Beit (Bao et al., 2021) and masked-autoencoder (He et al., 2022) in computer vision, among others.

**Quantization approaches**  Fan et al. (2020) propose Quant-noise, that mimicks quantization during training by introducing noise to a random subset of weights for each forward pass enabling high compression ratios while maintaining the original model performance. It uses the Straight-Through estimator (STE) (Bengio et al., 2013; Hinton, 2012) to compute the gradients. This training technique ensures that the model is pretrained to observe both the train-time (unquantized) and the inference-time (quantized) models. This ensures proper optimization, bypassing the flat gradient caused by quantization and reducing the discrepancy that results from the late quantization of the model weights.

**Sparsity by design**  Some works propose to enable sparsity directly in the architecture, for instance, the Mixture of Experts (MoE) or the Product-Key Memory (PKM). The PKM architecture (Lample et al., 2019) uses a memory layer for neural networks which enables the model to access a large learnable memory and thus, it enables long term memory capabilities. It leverages product quantization (PQ) (Jégou et al., 2011) by splitting the key in two parts and using each part in separate codebooks. The combination of each PQ index enables the model to access a larger memory space efficiently. At each forward pass, only a small subset of the memory is accessed, making it computationally efficient.

Mixture of Expert (MoE) models (Yang et al., 2024; Wei et al., 2024; DeepSeek-AI et al., 2024; Jiang et al., 2024) dynamically select and activate the relevant subset of parameters based on the characteristics of the input data. The MoE approach allows MoE models to expand their capacity without proportionally increasing computational complexity. See Mu & Lin (2025) for an overview of the MoE and references therein.

## 3  USING DIFFERENT ACTIVATIONS AT TRAIN AND TEST TIME

This section introduces two strategies for improving the optimization during pretraining using an optimization-compliant activation, while preparing the model to a potentially different activation at test time. First we introduce the `Swi+FT` fine-tuning approach. Then we introduce our Stochastic Activation `StochA`.

### 3.1  FINE-TUNING WITH RELU: SWI+FT

In the following, we use `SILU` and `RELU` as our training and inference activations. For reference, they are defined in $\mathbb{R} \to \mathbb{R}$ as:

$$\texttt{RELU}(x) = \max(x, 0) \qquad \texttt{SILU}(x) = x\sigma(x), \tag{1}$$

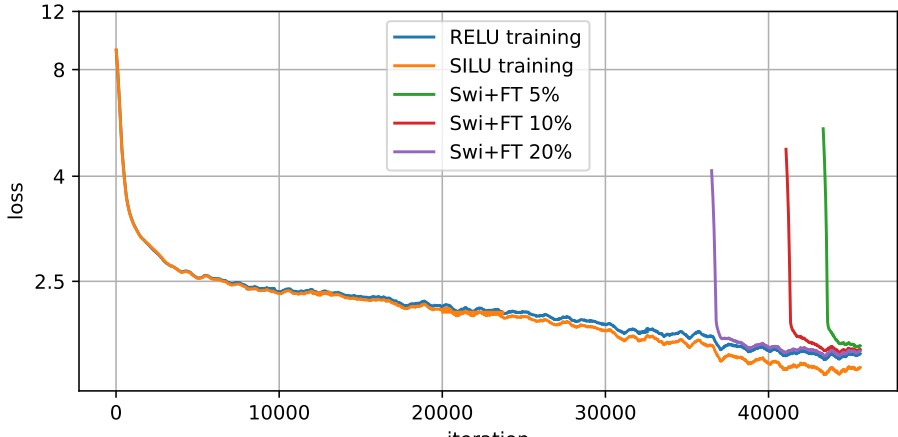

Figure 2: `Swi+FT`: Training loss. Most of the training is carried out with SILU, with $\alpha * 100\% = 5\%$, $10\%$ and $20\%$ of the final steps using RELU. Note the loss spike when we switch the activation. The model rapidly recovers and converges to a regime where `RELU` is performing well while providing sparsity.

where $\sigma(x) = 1/(1 + \exp(-x))$ is the sigmoid function. We choose these two activations because `SILU` is one of the best options in terms of accuracy, while `RELU` is simple and sparse. The two activations are also similar: same asymptotes at $-\infty$ and $+\infty$, and the same value at 0. `SILU` is differentiable twice (unlike `RELU`) and, interestingly, non-monotonous.

In our proposed approach, the training operates as follows:

- Most of the training steps (during a proportion $1 - \alpha$ of the total number of iterations) are carried out with a first activation that is deemed preferable for training. We typically employ `SILU` for this stage.
- We then switch the activation to that used for inference for the rest of the training.

We mostly set $\alpha = 0.05$ or $\alpha = 0.1$, which mean that only $5\%$ or $10\%$ of the training steps are carried out using the inference-time activation. We do not re-initialize the parameters of the optimizer when switching between activations, and similarly we do not use any warm-up. This does not disrupt the optimization because the `SILU` and `RELU` activations are relatively similar. We observe a spike in the loss at the time we change the activation, see Figure 2. However, the optimization rapidly recovers. In practice, the fine-tuning replaces the last iterations of the pretraining. The learning rate follows a cosine schedule which gradually reduces it to $1/100^{\text{th}}$ of its peak value. Therefore, at $5\%$ or $10\%$ of the end of the training, the learning rate is already $60\times$ or $29\times$ lower than its peak, which is compatible with a fine-tuning regime.

### 3.2 Stochastic activation: StochA

A stochastic function, parametrized by a random variable $\omega$, is a function

$$y = \Psi(x, \omega) \tag{2}$$

that maps inputs $x \in \mathbb{R}$ to output $y \in \mathbb{R}$ with randomness involved. The dependence on $\omega$ emphasizes that the outcome depends on an underlying probability space. In that sense, the function $\Psi(\cdot, \omega)$ is deterministic for each realization of $\omega$, but is stochastic overall. In particular, we consider the case depicted in Figure 1, where $\omega \sim \text{Bernoulli}(p)$ is a binary random variable parametrized by a parameter $p$: $\omega \in \{0, 1\}$ such that $\mathbb{P}(\omega = 1) = p$ and $\mathbb{P}(\omega = 0) = 1 - p$. In that case, the stochastic function $\Psi_p(\cdot)$ is defined such that

$$\text{if } x < 0, \ \Psi_p(x) = (1 - \omega) \times \text{RELU}(x) + \omega \times \text{SILU}(x), \tag{3}$$

which corresponds to randomly selecting between the `RELU` and `SILU` activations for $x < 0$. If $x \geq 0$ we choose $\Psi_p(x) = x$ or $\Psi_p(x) = \texttt{SILU}(x)$, see the baselines description below.

This strategy ensures that the network is compatible with two regimes. The first one, drawn with probability $1 - p$, is the inference-time mode, where we prepare the network to employ `RELU` during generation, in order to exhibit sparsity. The second mode aims to facilitate optimization during training. The choice of the `SILU` activation is motivated by the regular deterministic gated design by (Shazeer, 2020) adopted by most state-of-the-art LLMs.

**Notation** To specify an activation, we separately define the function for the positive and negative range of inputs. For example `R-S+` means that `RELU` is used for the negative range and `SILU` for the positive. When `StochA` is used, we indicate `[S|R]-S+`, which means that for the negative range, we sample `SILU` with probability $p$ and `RELU` with probability $1 - p$.

**Baselines** The two natural baselines are the deterministic functions `SILU` and `RELU`. We also introduce two non-stochastic baselines in order to disentangle the effect that could come from combining `SILU` and `RELU` separately in the positive and negative domain: these baselines are denoted by `S-R+` and `S-R+`.

|        | $x < 0$          | $x \geq 0$       |
| ------ | ---------------- | ---------------- |
| RELU   | $0$              | $x$              |
| SILU   | $x \cdot \sigma(x)$ | $x \cdot \sigma(x)$ |
| R-S+   | $0$              | $x \cdot \sigma(x)$ |
| S-R+   | $x \cdot \sigma(x)$ | $x$              |

**Discussion** The stochastic strategy resembles activation dropout (Srivastava et al., 2014), which can be regarded as a particular case of our method where one of the activations is the null function. However, the objective of dropout is to avoid overfitting. Our motivation is closer to Quantization-aware training (Jacob et al., 2017), more specifically, to the QuantNoise strategy of Fan et al. (2020), where the model is pretrained to observe both the train-time (unquantized) and the inference-time (quantized) models. In QuantNoise, using these two modes during training time ensure both the proper optimization, without suffering the flat gradient inherent to quantization, while reducing the discrepancy that results from the late-quantization of the model weights.

**Alternative construction of a stochastic activation.** An alternative construction is to randomly select between the identity function $x \mapsto x$ and the constant zero function $x \mapsto 0$ with a sigmoidal probability $\sigma(x)$. As a result, in expectation this function is given by

$$\mathbb{E}[\text{sa}(x)] = (1 - \sigma(x)) \cdot 0 + \sigma(x) \cdot x = \sigma(x) \cdot x, \tag{4}$$

where we recognize the $\texttt{SILU}(x)$ function. While the simplicity of this construction is mathematically appealing, our preliminary experiments revealed that it does not work very well.

### 3.3 Inference-time strategies and evaluation

At test time, we evaluate and analyze models trained with `Swi+FT` and/or `StochA` as follows:

**RELU at test time** This is how we can enable sparsity. The corresponding evaluations therefore measure the performance on benchmarks when using this activation at test time.

**Exploiting sparsity** On an input $x \in \mathbb{R}^D$, the gated FFN computes:

$$y = W_2 \times (\texttt{RELU}(W_1 \times x) \odot (W_3 \times x)) \text{ with } W_1, W_3 \in \mathbb{R}^{N \times D} \text{ and } W_2 \in \mathbb{R}^{D \times N}, \tag{5}$$

assuming column vectors and noting $\odot$ the element-wise multiplication. When the activation $\texttt{RELU}(W_1 \times x)$ has a fraction $s$ of zeros, the multiplications by $W_2$ and $W_3$ can exploit this sparsity: the baseline of $3ND$ FLOPS reduces to $(3 - 2s)ND$.

Note that exploiting the sparsity to increase the computational throughput is not straightforward. At training time, the runtime is dominated by matrix-matrix multiplications, where even a 90% sparsity rate is not guaranteed to yield efficiency gains. At inference time with one prompt at a time, the bottleneck is the memory access used during matrix-vector multiplications. When $W_2$ is stored by rows and $W_3$ by columns, the sparsity can be exploited to avoid a fraction $s$ of the memory reads, that are contiguous. This implementation nearly yields the expected speedup (see experimental section).

**Stochastic activation at test time**  The following only applies to the `StochA` strategy: we evaluate the performance by leveraging the randomness at test time, i.e., in this case, we do not use `RELU`. This choice is interesting for two reasons:

1. To quantify the effect of the activation discrepancy between train and test.
2. To generate multiple outputs from the same prompt with the randomness of `StochA`.

For the second usage, the standard way to generate multiple outputs from the same prompt is to replace the greedy decoding with a random sampling of the token from its probability distribution. This sampling can be tuned by setting a softmax temperature $T$ which adjusts between completely uniform sampling ($T \to \infty$) and strict maximum sampling ($T \to 0$). In both cases, we keep the one generated output with the highest normalized log likelihoods, *i.e.*, the per-token average log-likelihood, as predicted by the model.

## 4 EXPERIMENTS WITH LARGE LANGUAGE MODELS

### 4.1 EXPERIMENTAL SETTING

**Model architecture**  We train dense decoder-only models. The transformer blocks use grouped-query attention (Ainslie et al., 2023). These models use RMSNorm (Zhang & Sennrich, 2019) with prenormalization, rotary positional positional encoding (RoPE) (Su et al., 2021) with $\theta = 500000$ and train with document causal masking. We use the `SILU` activation (Shazeer, 2020) for the `SILU` baseline. The structure of our `LM1.5B` and `LM3B` models is detailed in Table 4 in Appendix A.

**Training hyper-parameters**  We train the models with AdamW optimizer (Loshchilov & Hutter, 2017) with $\beta_1 = 0.9$, $\beta_2 = 0.95$, learning rate of $lr = 3 \times 10^{-3}$, weight decay of 0.1, and gradient clipping at norm 1.0. After 2000 steps of linear warm-up, we use a cosine decay learning rate schedule with peak learning rate $8 \times 10^{-4}$ and decay by a factor of $1/100$ over the training horizon.

**Tokenizer**  We use the Llama3 (Dubey et al., 2024) tokenizer, which is a fast Byte-Pair Encoding tokenizer implemented with TikToken.2 The vocabulary contains 128 000 regular tokens as well as 256 reserved tokens.

**Pre-training**  We pre-train the `LM1.5B` and `LM3B` models with 47B and 80B tokens, respectively, from a diverse collection of mostly English natural language and coding data. We use a batch size of 1M tokens and a context length of 8192 tokens.

**Evaluation Benchmarks**  We employ two types of benchmarks for zero or few-shot evaluation, which we describe in more detail in Appendix F and Table 8. The first type is code generation tasks: HumanEval+ (Liu et al., 2023) and MBPP (Chen et al., 2021). The second type consists of common sense and general reasoning: HellaSWAG(Zellers et al., 2019), ARC(Clark et al., 2018), PIQA (Bisk et al., 2020), OBQA (Mihaylov et al., 2018), WinoGrande (Sakaguchi et al., 2020), NQ (Kwiatkowski et al., 2019), RACE (Lai et al., 2017), TQA (Joshi et al., 2017) and GSM8K (Cobbe et al., 2021).

### 4.2 PERFORMANCE ANALYSIS OF SWI+FT AND STOCHA WITH RELU AT INFERENCE TIME

In this section we analyze the effect of our proposal when using `RELU` at test time. In Appendix B, we provide a complementary analysis of the sparsity. Depending on the setting, the average rate of 0s can be higher than 90%, when using the `RELU` at test time.

**Cross-entropy performance**  Table 1 provides the impact on the training and validation losses of multiple choices with the `LM1.5B` and `LM3B` models. We observe that the training loss using stochastic activation at train time is lower than that obtained with `RELU`. However the validation entropy is not competitive per se, due to the remaining train-inference

| Activation | Training activation | | | | | LM1.5B | | | LM3B | | |
| --- | --- | --- | --- | --- | --- | --- | --- | --- | --- | --- | --- |
| | $x < 0$ | | $x > 0$ | $p$ | | train | val | val | train | val | val |
| | $p$ | $1-p$ | | | Swi+FT | | RELU | StochA | | RELU | StochA |
| SILU | SILU | | SILU | - | ✗ | 2.105 | 2.122* | | 1.966 | 1.974* | |
| RELU | RELU | | RELU | - | ✗ | 2.140 | 2.161 | | 2.027 | 2.043 | |
| S-R+ | SILU | | RELU | - | ✗ | 2.101 | 2.124* | | 1.970 | 1.980* | |
| R-S+ | RELU | | SILU | - | ✗ | 2.123 | 2.151* | | 2.016 | 2.033* | |
| [S\|R]-R+ | SILU | RELU | RELU | 0.3 | ✗ | 2.120 | 2.363 | 2.146 | 1.993 | 2.257 | 2.006 |
| [S\|R]-R+ | SILU | RELU | RELU | 0.5 | ✗ | 2.120 | 2.507 | 2.145 | 1.990 | 2.889 | 1.999 |
| [S\|R]-S+ | SILU | RELU | SILU | 0.3 | ✗ | 2.115 | 2.305 | 2.143 | 1.987 | 2.257 | 1.996 |
| [S\|R]-S+ | SILU | RELU | SILU | 0.5 | ✗ | 2.115 | 2.530 | 2.143 | 1.984 | 2.753 | 1.995 |
| [S\|R]-R+ | SILU | RELU | RELU | 0.3 | ✓ | 2.123 | 2.141 | 2.251 | 1.988 | 1.998 | 2.177 |
| [S\|R]-R+ | SILU | RELU | RELU | 0.5 | ✓ | 2.129 | 2.148 | 2.307 | 1.989 | 2.002 | 2.306 |
| [S\|R]-S+ | SILU | RELU | SILU | 0.3 | ✓ | 2.120 | 2.138 | 2.221 | 1.982 | 1.992 | 2.103 |
| [S\|R]-S+ | SILU | RELU | SILU | 0.5 | ✓ | 2.125 | 2.144 | 2.301 | 1.985 | 1.994 | 2.234 |

Table 1: The train loss is computed over the last 500 steps of the training of `LM1.5B`, the val loss is measured after training, on a different set of text and code, using the `RELU` activation* or `StochA`, i.e. the same activation used at train time (possibly deterministic). If `Swi+FT` is enabled, we switch to `RELU` for the last 5% steps. *: for the deterministic baselines `SILU`, `S-R+` and `S+R-`, we do the inference with the same activation used at train-time (not `RELU`).

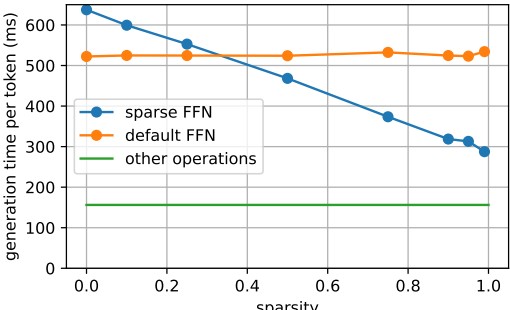

Figure 3: Total inference time for 1 token on CPU, as a function of the activation sparsity, with a `LM3B` model trained with `Swi+FT`. The "other operations" include the attention layers (they are not dominant because the generation is limited to 200 tokens), the normalization and the execution overheads. At 90% sparsity the speedup is ×1.65. The timings are measured on a single core of a Xeon 8462Y+ machine.

discrepancy of activation. This is solved by `Swi+FT`: switching to the `RELU` activation function and fine-tuning for the last 5% or 10% steps of the training steps drastically boosts the test-time inference. These results outperform the results obtained with regular `RELU` training, while using the same activation at test time.

**Fast inference with `RELU` sparsity** The activation sparsity can be exploited to avoid fetching 90% of the matrices $W_1$ and $W_3$ of the FFN (see Eq. 5 and detailed sparsity rates in Appendix B). As a consequence, we observe a direct benefit on CPU: Figure 3 shows that 90% sparsity directly translates into a 65% speedup. On GPU, the additional challenge is to make the computation sufficiently predictable to balance the load between CUDA threads.

**Absence of dead neurons with `Swi+FT`** Figure 7(*left*) shows a sudden jump in the `RELU` CDF, which indicates there are many activations that are exactly zero *on input* of `RELU`. The `RELU` plot in Figure 7(*right*) shows that a large fraction of the rows of the linear layer $W_1$ (Eq 5) are near zero (*i.e.* their norm is less than 1/1000 the average row norm of the matrix). This means that these rows are unused: they are "dead neurons". In contrast, the `Swi+FT` plot in Figure 7(*right*) we observe that using `SILU` at pre-training prevents these dead neurons appearing. This explains why our approach obtains significantly better results with `RELU` than the vanilla training of the model using the `RELU` from scratch.

**Complementarity between `StochA` and `Swi+FT` when fine-tuned with `RELU`** Figure 4 shows the training loss when using `StochA` and `Swi+FT` jointly. When switching the activation to `RELU`, we observe a spike in the loss, but the optimization rapidly recovers, and converges to a model that has better performance than the one trained with `RELU` from scratch. In contrast, in the case where we employ `Swi+FT` alone (Figure 2), fine-tuning

with `RELU` after pretraining with `SILU` is not enough to obtain performance improvements. As a consequence, the `StochA` and `Swi+FT` approaches are complementary.

**Impact of fine-tuning with `RELU` for the last $\alpha * 100\%$ steps**  In Figure 4, we compare the final loss reported for different values of $\alpha$. In that case, setting $\alpha = 0.1$ is a good trade-off since the corresponding loss is lower than the `RELU` loss.

**Continuous pre-training with `StochA` and `Swi+FT`**  Starting from a pre-trained `LM1.5B` model with `SILU`, we can leverage either `StochA` or `Swi+FT` methods for continuous pretraining (CPT), see Appendix D for experimental details. In such case, the best strategy is to use `Swi+FT`, i.e., simply fine-tune with `RELU` the model initially trained with `SILU` to obtain the best trade-off between sparsity and loss (Table 6). This pattern is the opposite of what we observe for LMs trained from scratch, where the `StochA` and `StochA +Swi+FT` offer the best trade-off between sparsity and loss (Table 1).

**Generalizability of `StochA` and `Swi+FT` for any pair of activations**  In Appendix E, we explain how our methods can be generalized for any pair of activation functions and include an illustrative example with the (`TANH`,`RELU`) activations.

### 4.3  PERFORMANCE ON DOWNSTREAM TASKS

**Detailed results per benchmark**  Table 2 reports the results for standard code generation, common sense and general reasoning benchmarks detailed in Appendix F. We consider multiple `StochA` models using few-shot or zero shot prompting, see Table 8 in Appendix F for more details. The model with `SILU` (topline) is significantly better that a regular model with `RELU`. However, our models trained with `StochA` are slightly better or on par with `SILU`: either the model fine-tuned with `Swi+FT` and using `RELU` at inference time, or even the model that uses `StochA` at test time.

**Performance when varying $\alpha$ with `Swi+FT`**  Figure 5 shows that we can slightly surpass the `SILU` baseline average performance if we first use a stochastic activation function during the `LM1.5B` model training and then switch to the `RELU` activation for the last $\alpha * 100\%$ of the training steps, for $\alpha \in \{0.05, 0.1, 0.2\}$. The best performance is obtained with $\alpha = 0.05$ or $\alpha = 0.1$ for the `LM1.5B` model.

### 4.4  EXPLOITING `STOCHA` AT TEST TIME

**Effectiveness of `StochA` a test time**  In Table 1, in addition to the results with `RELU` at test time, we also report the train and validation losses obtained when employing `StochA` at test time. We observe that (1) using stochastic activations for inference works surprisingly well in spite of the randomness. The results are between `RELU` and `SILU` in most configurations; (2) When using `StochA` at test time, there is no need to fine-tune the model with `Swi+FT`. This is expected since this strategy is intended to decrease the discrepancy with the

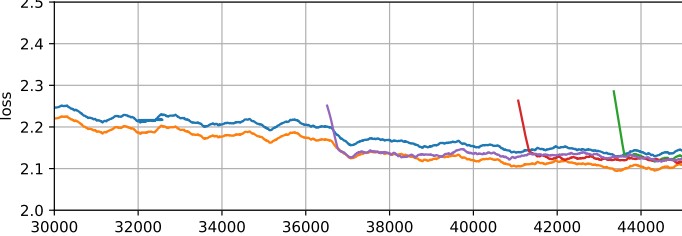

| $\alpha$ | final loss | |
| --- | --- | --- |
| | train | val (`RELU`) |
| 0 | 2.115 | 2.530 |
| 0.05 | 2.125 | 2.144 |
| 0.10 | 2.119 | 2.145 |
| 0.20 | 2.126 | 2.155 |

Figure 4: Training loss with `Swi+FT` and `StochA`: `[S|R]-S+` activation with $p = 0.5$ for $\alpha*100\%$=5%, 10% and 20%, relative to `RELU` and `SILU`. This shows that the `Swi+FT` strategy needs to be combined with `StochA` to provide good models operating with `RELU` compared to finetuning `SILU` with `RELU` alone (Figure 2). This plot this is zoomed in relative to Figure 2.

| ↓ **Benchmark/metric** | | | | | LM1.5B | | | | | | LM3B | | |
|---|---|---|---|---|---|---|---|---|---|---|---|---|---|
| | | (a) baselines | | (b) Swi+FT [S\|R]-S+ | (c) StochA [S\|R]-S+ | | (a) baselines | | (b) Swi+FT [S\|R]-S+ | (c) StochA [S\|R]-S+ | | | |
| train activation → inference activation → | | SILU SILU | RELU RELU | RELU | [S\|R]-S+ | | SILU SILU | RELU RELU | RELU | [S\|R]-S+ | | | |
| hellaswag/acc_char | | 0.585 | 0.561 | 0.574 | 0.576 | | 0.684 | 0.633 | 0.671 | 0.678 | | | |
| winogrande/acc_char | | 0.593 | 0.571 | 0.568 | 0.568 | | 0.657 | 0.615 | 0.630 | 0.620 | | | |
| arc_easy/acc_char | | 0.568 | 0.562 | 0.600 | 0.562 | | 0.675 | 0.642 | 0.679 | 0.671 | | | |
| arc_challenge/acc_char | | 0.313 | 0.286 | 0.331 | 0.314 | | 0.390 | 0.348 | 0.396 | 0.376 | | | |
| piqa/acc_char | | 0.732 | 0.720 | 0.724 | 0.720 | | 0.767 | 0.751 | 0.765 | 0.761 | | | |
| obqa/acc_char | | 0.346 | 0.340 | 0.378 | 0.340 | | 0.390 | 0.380 | 0.384 | 0.408 | | | |
| race.middle/acc_char | | 0.518 | 0.516 | 0.509 | 0.498 | | 0.565 | 0.538 | 0.559 | 0.549 | | | |
| race.high/acc_char | | 0.382 | 0.379 | 0.372 | 0.375 | | 0.414 | 0.402 | 0.407 | 0.416 | | | |
| human_eval_plus/pass@1 | | 0.073 | 0.067 | 0.049 | 0.055 | | 0.128 | 0.110 | 0.128 | 0.116 | | | |
| mbpp/compiles@1 | | 0.978 | 0.970 | 0.960 | 0.980 | | 0.992 | 0.980 | 0.990 | 0.982 | | | |
| tqa/f1 | | 0.243 | 0.217 | 0.229 | 0.232 | | 0.351 | 0.293 | 0.327 | 0.342 | | | |
| nq/f1 | | 0.123 | 0.107 | 0.121 | 0.113 | | 0.169 | 0.146 | 0.170 | 0.145 | | | |
| average performance | | **0.454** | 0.441 | 0.451 | 0.444 | | **0.515** | 0.486 | 0.509 | 0.505 | | | |

Table 2: Performance per benchmark of (a) RELU and SILU baselines for LM1.5B and LM3B compared to (b) models with StochA +Swi+FT at train time and RELU at test time, and (c) models with StochA at train and test time. We use the model with the best perplexity on val namely $p = 0.3, \alpha = 0.05$ for Swi+FT and $p = 0.5$ for StochA. The average performance for the topline (SILU) is in bold, second best (StochA +Swi+FT) is underlined.

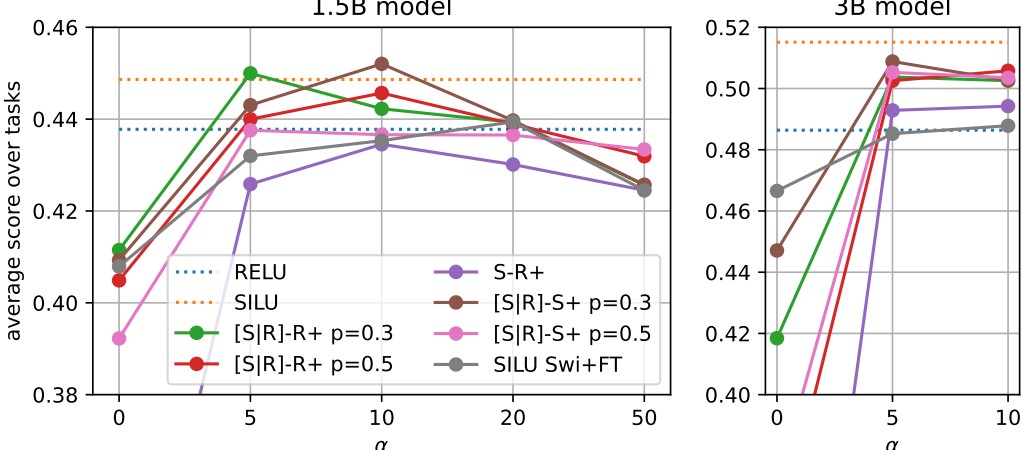

Figure 5: **Swi+FT: analysis of the fine-tuning rate** $\alpha$. Average performance over the benchmarks as a function of the percentage $\alpha$ of steps for which we switch to the RELU activation at the end of training. We use RELU at inference time.

test-time activation choice. Table 3 shows that the average benchmark performance generally increases when the stochastic mix approaches SILU. Therefore, StochA is primarily useful as a way to generate multiple outputs for the same prompt.

| $p$ | | LM1.5B | LM3B |
|---|---|---|---|
| 0 | (R-S+) | 0.215 | 0.222 |
| 0.3 | | 0.443 | 0.504 |
| 0.5 | | 0.426 | 0.505 |
| 0.7 | | 0.453 | 0.495 |
| 1.0 | (SILU) | 0.454 | 0.515 |

Table 3: **StochA**: Impact on benchmarks performance (avg) as a function of the StochA $p$ for [S\|R]-S+ used at test time. The case $p = 0$ corresponds to R-S+ while $p = 1$ corresponds to the baseline SILU. The performance increases with more SILU in the mix. However, the stochasticity can be used to increase the generation diversity.

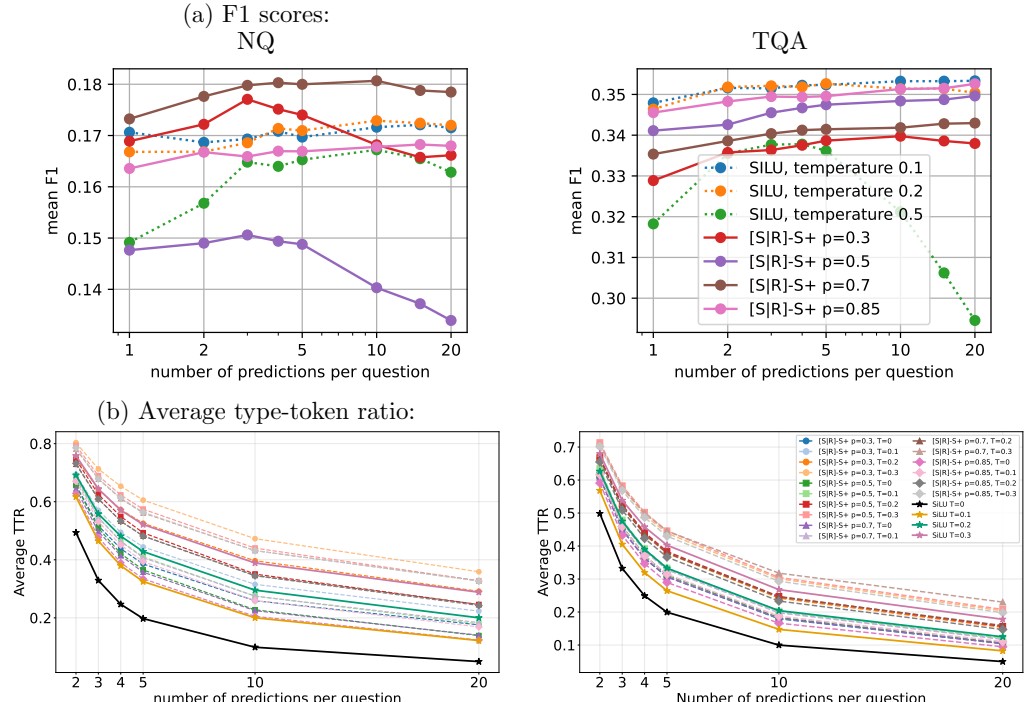

Figure 6: Comparison of diverse response generation methods in Q&A benchmarks. (a) F1 scores when varying the temperature with `SILU` or `[S|R]-S+` $p$ with temp$= 0$, the generations are scored by normalized log likelihoods. (b) TTR when varying the temperature in temperature sampling with `SILU` vs with `[S|R]-S+` for different values of $p$.

**Diversity of generations ablation** Figure 6 shows that leveraging the `StochA` activations stochasticity (combined with standard temperature sampling) generates more diverse answers compared to using `SILU` with standard temperature sampling. See Appendix C for the experimental setup and Appendix G for qualitative examples of such generations.

In Figure 6 (a), the curves are increasing, which means that (1) we obtain diverse generations and (2) that the normalized log-likelihood is a suitable scoring function. Specifically, for the NQ task, the `StochA` activation with $p = 0.7$ and temperature zero yields consistently higher performance than standard temperature sampling with the `SILU` activation (for temperatures in $\{0.1, 0.2, 0.5\}$). The `StochA` activation with $p = 0.3$ is also competitive for the number of generations up to five. In contrast, the results are subpar for the TQA task: standard temperature sampling with the `SILU` activation yields higher performances for all number of generations for temperatures in $\{0.1, 0.2\}$.

In Figure 6 (b), we observe that all the models that use `StochA` activation functions have consistently higher average TTR for both tasks, for each temperature in $\{0, 0.1, 0.2, 0.3\}$ compared to standard temperature sampling with `SILU`. This finding further confirms that the text diversity of generations is much better for these models.

## 5 CONCLUSION

This paper introduces a novel stochastic activation that preserves the performance of a non-sparse activation, such as `SILU`, while better adjusting to the behavior of a sparse one, such as `RELU`, at test time. This improves the inference times for the FFN layers of a transformer, translating into a speedup of typically $\times 1.65$ for the FFN processing on CPUs while almost preserving the accuracy of the non-sparse `SILU` activation. Finally, we explore how stochastic activations can be leveraged at test time to improve diversity in model generations.

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

# Appendix

In this appendix, we add some details that did not fit in the main paper. Appendix A gives architectural details for the models used in the paper. Appendix B gives additional details of the sparsity produced by the models that use the `StochA` and `Swi+FT` activations. Appendix C provides the experimental details for the diversity of generations ablation. Appendix D gives some details for the CPT experiment with `StochA` and `Swi+FT`. Appendix E discusses how to generalize the `StochA` and `Swi+FT` methods for any pair of non-sparse and sparse activations. Appendix F gives the references for all the benchmarks used in the experiments. The final Appendix G gives some examples of multiple answers generated by `StochA`.

## A  Architecture detail

The model architectures we use in the paper are loosely inspired by Llama 3. Table 4 summarizes their main parameters.

| Parameter | LM1.5B | LM3B |
|---|---|---|
| Number of parameters | 1.5B | 3B |
| Layers | 28 | 36 |
| Hidden dimension | 1536 | 2048 |
| Intermediate dimension | 8960 | 11008 |
| Number of attention heads | 12 | 16 |
| Number of key-value heads | 2 | 2 |

Table 4: `LM1.5B` and `LM3B` model parameters

## B  Sparsity Analysis

In Table 5, we report the sparsity ratios resulting from the gating in the FFN layer, when using the following activations: `RELU`, `SILU`, `Swi+FT` fine-tuning, and when using `StochA` with `Swi+FT` for varying values $p$. In all cases, except for the `SILU` baseline, we use `RELU` at test time. We observe that `Swi+FT` fine-tuning with `RELU` brings back a lot of sparsity (80%), yet the model is not competitive. The combination of `StochA` with `Swi+FT` leads to models offering a high sparsity degree by using `RELU` at test time and simultaneously achieving competitive performance, see Tables 1 and 2.

In Figure 7 (*left*), we observe that $P(X \leq 0)$ is the highest for `RELU` since most of its mass is concentrated around zero as in Table 5. For $x < 0$, the probability $P(X \leq x)$ is lower for `SILU` than for both `Swi+FT` and when we combine `StochA` with `Swi+FT` for different values of $p$. This shows that combining `StochA` with `Swi+FT` enables to control how much mass we assign to negative values of $x$ depending on $p$.

| | train | inference | Swi+FT | StochA: p | Swi+FT: $\alpha$ | sparsity (%) |
|---|---|---|---|---|---|---|
| baselines | SILU | SILU | ✗ | - | | 0.0002 |
| | RELU | RELU | ✗ | - | | 94.8 |
| Swi+FT | SILU | RELU | ✓ | - | 0.05 | 79.9 |
| StochA | [S\|R]-S+ | RELU | ✓ | 0.3 | 0.05 | 88.5 |
| +Swi+FT | [S\|R]-S+ | RELU | ✓ | 0.5 | 0.05 | 86.5 |
| | [S\|R]-S+ | RELU | ✓ | 0.7 | 0.05 | 84.6 |

Table 5: We report the rate of the 0-valued activation after the `SILU` (not sparse) and `RELU` activation, when training with the `LM1.5B` model with the following activations: `RELU`, `SILU`, `Swi+FT` fine-tuning, and when using `StochA` with `Swi+FT` for varying values of $p$.

Pre-activation inputs distribution      Presence of dead neurons in the $W_1$ layer

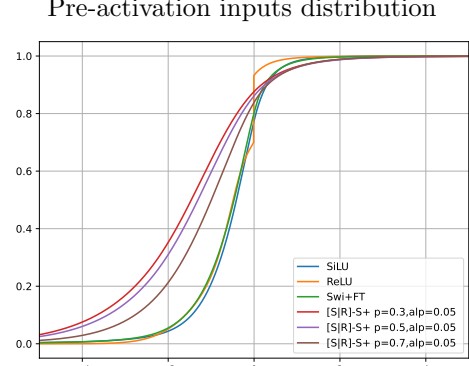
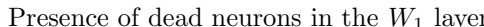
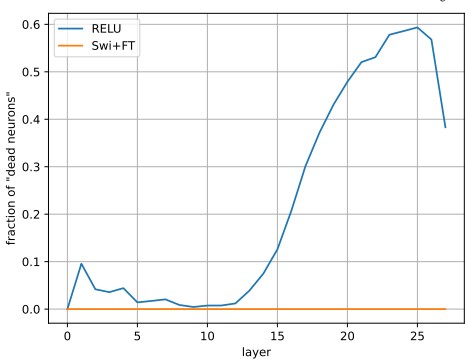

Figure 7: *Left:* empirical cumulative distribution functions for the inputs to the activation functions when training the `LM1.5B` model with one of the following activations: `SILU`, `RELU`, `Swi+FT` fine-tuning and the combination of `StochA` with `Swi+FT` for different values of $p$. *Right:* fraction of near-zero rows of the $W_1$ matrix, indicating useless neurons for two types of activations.

## C  DIVERSITY OF GENERATIONS ABLATION EXPERIMENTAL SETUP

We use the `LM3B` models trained with the `StochA` activation `[S|R]-S+` with $p$ in $\{0.3, 0.5, 0.7, 0.85\}$. At inference time, the probability $p$ is set to be the same as the one during training. We generate twenty answers for the NQ and TQA generation tasks, compute the Ff1 and Type-token ratio (TTR) metrics of $n$ generated answers (samples) and average over all datapoints for a given number of samples $n$ in $\{2, 3, 4, 5, 10, 20\}$.

In Figure 6 (b) we report the type-token ratio metric (Johnson, 1944), a simple and widely used metric in linguistics and natural language processing to measure lexical diversity in a text. In this context, a *type* consists of a unique word in the text and a *token* consists of any word occurrence in the text (including repetitions). The Type-Token Ratio is then *TTR=Number of Types / Number of Tokens*.

## D  CONTINUOUS PRETRAINING `LM1.5B` MODELS

We can leverage the `StochA` and `Swi+FT` methods for continuous pretraining (CPT) language models, besides using these activations to train LMs from scratch. In this experiment, we start from the `LM1.5B` models pretrained with the `SILU` activation. During CPT, we use i) `Swi+FT` ii) `StochA` with $p = 0.3$ or $p = 0.5$ or iii) `StochA` and `Swi+FT` with $p = 0.5$ and $\alpha = 0.1$. We use the same number of steps used in the pretraining experiments: 45,629 steps. We compare the training losses against the baseline model that keeps `SILU` during CPT.

|  | pretrain | CPT | inference | `StochA`: p | sparsity (%) | train loss |
|---|---|---|---|---|---|---|
| baseline | SILU | SILU | SILU | - | 0.0002 | **2.076** |
| Swi+FT | SILU | RELU | RELU | - | **90.96** | 2.086 |
| StochA | SILU | StochA | RELU | 0.5 | 41.02 | 2.095 |
|  | SILU | StochA | RELU | 0.3 | 57.93 | 2.095 |
| StochA +Swi+FT | SILU | StochA +Swi+FT | RELU | 0.5 | 81.61 | 2.096 |

Table 6: We report the rate of the 0-valued activations and training losses when doing CPT with the `LM1.5B` model with the following activations: `RELU`, `SILU`, `Swi+FT`,`StochA` and when using `StochA` with `Swi+FT` for varying values $p$.

| | Training activation | | | | | LM1.5B | | |
|---|---|---|---|---|---|---|---|---|
| | $x < 0$ | | $x > 0$ | $p$ | | train | val | val |
| Activation | $p$ | $1-p$ | | | Swi+FT | | RELU | StochA |
| TANH | TANH | | TANH | - | ✗ | 2.133 | 2.155 | |
| RELU | RELU | | RELU | - | ✗ | 2.140 | 2.162 | |
| Tanh+FT | TANH | | TANH | - | ✓ | 2.193 | 2.224 | |
| [T|R]-T+ | TANH | RELU | TANH | 0.3 | ✗ | 2.368 | 10.803 | 2.421 |
| [T|R]-T+ | TANH | RELU | TANH | 0.5 | ✗ | 2.314 | 10.997 | 2.359 |

Table 7: Losses: train is computed over the last 500 steps of the training loss of LM1.5B, val is measured after training on a different set of text and code using the RELU activation* or StochA, i.e., the same activation used at train time (possibly deterministic). If Swi+FT is enabled, we switch to RELU for the last 10% steps.

## E  STochA AND Swi+FT FOR (TANH, RELU) ACTIVATIONS

Our StochA and Swi+FT methods can be generalized, starting with any pair of non-sparse, sparse activations, in the following way:

1. StochA: the non-sparse (activation which takes non-zero values on the negative side) and sparse activation pair can be used as a stochastic activation function that takes the form of either of the activations on the positive side, and, on the negative side, it either takes the non-zero value of the non-sparse variant with probability $p$ or is set to zero with probability $1-p$.

2. Swi+FT: we start with an LM with a non-sparse activation and train it for $(1-\alpha)\%$ steps. Succesively, we finetune the LM with the sparse activation for $\alpha\%$ of the final steps.

To illustrate how the StochA and Swi+FT methods are generalizable, we use the pair of (TANH, RELU) activations. We pretrain a LM1.5B with TANH as the non-sparse activation and RELU as the sparse one for 1) and 2). We compare these versus the losses to the corresponding baselines: using only the TANH or only the RELU activations, respectively. We denote the activation for 1) as Tanh+FT with parameter $\alpha$ and for 2), with [T|R]-T+ with parameter $p$.

We observe that there is barely any gap between the train loss of the non-sparse (TANH, bolded) and the sparse (RELU, in red) activations, hence, there is no trade-off between sparsity and performance, in contrast to the (SILU, RELU) case.

## F  BENCHMARKS

**Code generation**  We use two benchmarks that evaluate the code generation capabilities of AI models: HumanEval+ and MBPP.

- The HumanEval+ (Liu et al., 2023) benchmark is an extension of HumanEval (Chen et al., 2021), which is designed to evaluate the functional correctness of code generated by AI models.

- MBPP (Austin et al., 2021) is designed to evaluate the code generation abilities of AI models, particularly for Python programming tasks.

**Common sense and general reasoning**  We use benchmarks consisting of question-answer or multiple-choice questions designed to evaluate the commonsense reasoning abilities of AI models, particularly in the context of natural language understanding: HellaSWAG, ARC, PIQA, OBQA, Winogrande, NaturalQuestions, RACE, TQA and GSM8K.

- HellaSWAG (Zellers et al., 2019) consists of multiple-choice questions where each question contains a short context (a sentence or paragraph) followed by four possible continuations. Only one continuation is correct and makes sense given the context.

- The AI Reasoning Challenge (ARC) (Clark et al., 2018) benchmark consists of multiple-choice science questions typically found in elementary and middle school exams. The ARC questions require a mix of factual knowledge, commonsense reasoning, and multi-step inference.

- The Physical Interaction Question-Answering (PIQA) (Bisk et al., 2020) benchmark consists of multiple-choice questions about how to accomplish simple physical tasks. Each question presents a short scenario and two possible solutions; only one is physically plausible.

- The OpenBook QA (OBQA) (Mihaylov et al., 2018) benchmark consists of 6,000 multiple-choice questions based on elementary science facts. Each question is designed to require combining a provided "open book" science fact with additional commonsense or general knowledge.

- WinoGrande (Sakaguchi et al., 2020) benchmark consists of multiple-choice questions. Each question presents a sentence with a pronoun and two possible antecedents; the task is to choose the correct referent for the pronoun.

- Natural Questions (NQ) (Kwiatkowski et al., 2019) benchmark is designed to evaluate the ability of an AI model to answer real user questions using information from Wikipedia.

- The Reading Comprehension from Examinations (RACE) (Lai et al., 2017) benchmark consists of passages and multiple-choice questions to assess how well AI models can comprehend and reason about written passages.

- The Trivia QA benchmark (TQA) (Joshi et al., 2017) is a reading comprehension dataset that pairs trivia questions with evidence documents from which answers can be derived.

- The Grade School Math 8k (GSM8k) (Cobbe et al., 2021) benchmark is a dataset of 8,500 high-quality, linguistically diverse grade school math word problems. The benchmark is designed to test multi-step mathematical reasoning capabilities in language models.

| Benchmark | Metric | Few Shot | Type |
|---|---|---|---|
| hellaswag | acc_char | 0 | choice |
| winogrande | acc_char | 0 | choice |
| arc_easy | acc_char | 0 | choice |
| arc_challenge | acc_char | 0 | choice |
| piqa | acc_char | 0 | choice |
| obqa | acc_char | 0 | choice |
| race.middle | acc_char | 0 | choice |
| race.high | acc_char | 0 | choice |
| human_eval_plus | pass@1 | 0 | generation |
| mbpp | compiles@1 | 3 | generation |
| tqa | f1 | 5 | generation |
| nq | f1 | 5 | generation |

Table 8: Summary of benchmarks used for evaluation. We few-shot prompt the `LM1.5B` and `LM3B` models. The metric depends on the benchmark as well as the average performance.

## G  EXAMPLES OF VARIOUS SEQUENCES GENERATED WITH STOCHA

Using a pre-trained `LM3B` model with `StochA`, we can use the `StochA` activation at test time to generate multiple predictions by leveraging the randomness from the activation function. We provide two examples of the generations obtained using questions from the TQA benchmark (Joshi et al., 2017):

| question and accepted answers | generated answers | number of times | NLL score | F1 score |
|---|---|---|---|---|
| **When did ibuprofen become available over the counter?** A: 1983 A: 1984 | 1988 | 4 | 1.002 | 0 |
| | 1982 | 11 | 1.004 | 0 |
| | 1980 | 1 | 1.030 | 0 |
| | 1995 | 2 | 1.019 | 0 |
| | 2001 | 1 | 1.017 | 0 |
| | 1983 | 1 | 1.024 | 1 |
| **Who played michael jackson in jackson 5 movie?** A:alex burrall A:abolade david olatunde A:wylie draper A:jason weaver | Michael Jackson | 19 | 0.948 | 0 |
| | Michael jackson | 1 | 0.792 | 0 |
| **Where is the meridian that is opposite the prime meridian located?** A:180th meridian A:antimeridian | Greenwich | 9 | 0.896 | 0 |
| | Greenwich, England | 11 | 0.551 | 0 |
| **The cold dry winds that blow over northern india in winter are called?** A:northeast monsoon A:retreating monsoon A:northeast monsoon or retreating monsoon | Siberian | 7 | 1.099 | 0 |
| | Siberian winds | 6 | 0.970 | 0 |
| | Ganges | 1 | 1.617 | 0 |
| | monsoon | 2 | 1.051 | 2/3 |
| | katabatic winds | 3 | 0.738 | 0 |
| | Ganga | 1 | 1.643 | 0 |
| **Where can you find dna in the body?** A:chromosomes in cell A:inside cell nucleus | Blood | 6 | 1.262 | 0 |
| | Hair | 12 | 1.393 | 0 |
| | Mitochondria | 1 | 0.700 | 0 |
| | In the nucleus | 1 | 1.437 | 2/5 |
| **Who is often associated with printing the first book using moveable type in germany?** A:johannes gutenberg | Johannes Gutenberg | 20 | 0.205 | 1 |
| **Who won the womens 2017 ncaa basketball tournament?** A:south carolina | Kentucky | 15 | 1.852 | 0 |
| | Kentucky Wildcats | 1 | 1.304 | 0 |
| | North Carolina | 4 | 1.342 | 1/2 |
| **Country with most olympic gold medals all time?** A:united states | USA | 14 | 0.818 | 0 |
| | United States | 6 | 0.614 | 1 |
| **The atomic number of indium which belongs to 5th period is?** A:49 | 5 | 14 | 0.579 | 0 |
| | 49 | 5 | 0.650 | 1 |
| | 84 | 1 | 0.695 | 0 |
| **Who appoints the members of the board of governors of the federal reserve?** A:president | The president | 16 | 0.602 | 1 |
| | The president of the United States | 4 | 0.431 | 2/5 |
| **What age do you need to be to buy a bb gun?** A:18 | 14 | 17 | 0.660 | 0 |
| | 10 | 2 | 0.680 | 0 |
| | 18 | 1 | 0.728 | 1 |
| **What genre is the magic tree house books?** A:childrens historical fantasy | Fantasy | 14 | 1.038 | 1/2 |
| | Children's fiction | 1 | 1.051 | 2/5 |
| | Children's | 3 | 1.245 | 1/2 |
| | Children's books | 2 | 1.067 | 2/5 |
| **What is the name of the skin between your nostrils?** A:nasal septum A:septum | Nasal septum | 17 | 0.707 | 1 |
| | Nasion | 3 | 1.224 | 0 |

Table 9: Example generations. For each question, we indicate the ground-truth answers (from the dataset). We generate 20 answers per question with [S|R]-S+, $p=0.7$. We list the de-duplicated answers, with the NLL score (used to sort the results) and the F1 score (used to evaluate the result, it is computed as the intersection of bags-of-words).

