# OpenReview forum: "Stochastic activations"
_ICLR.cc/2026/Conference — Submitted to ICLR 2026_

### Official Review · Reviewer_q6x3 · 2025-10-25

**Soundness:** 3
**Presentation:** 3
**Contribution:** 2
**Rating:** 4
**Confidence:** 3

**Summary:**

This paper proposes training LLMs with stochastic activations to strike a balance between performance sparsity/efficiency. The performance seems promising but it is unclear how much practical benefit this can have on GPUs.

**Strengths:**

1. Ideas and experiments are clear and easy to understand.
2. Training with stochastic activations is shown to have performance on par with SiLU activations, resolving a major weakness of ReLUs.
3. Experiments are thorough.

**Weaknesses:**

1. Practical use case for this method is unclear since GPU acceleration is not demonstrated.
2. The generalizability is unclear. Will this setup work with pairs of activations that are less similar (e.g., ReLU + tanh)? What about non-ReLU sparse variants (e.g., soft/hard thresholding)?
3. Lack of structured sparsity baselines.

**Questions:**

1. Line 143: Unremoved comment
2. Hard to see what is going on at the tail in figure 4. Could you shrink the y-axis range?
3. Can you comment on when/how your method is preferred over post-training sparsity (e.g., [1]) and inference-time sparsity algorithms (e.g. [2,3])?

[1] Zhang, et al., MoEfication: Transformer Feed-forward Layers are Mixtures of Experts, 2021.
[2] Dong, et al., Prompt-prompted Adaptive Structured Pruning for Efficient LLM Generation, 2024.
[3] Lee, et al., Cats: Contextually-aware thresholding for sparsity in large language models, 2024.

---

> ### Author Response · Authors · 2025-11-25
>
> We thank the reviewer for their comments.
>
> *The generalizability is unclear. Will this setup work with pairs of activations that are less similar (e.g., ReLU + tanh)?*
>
> Firstly, the reason why we choose the SiLU activation is because language models (LMs) use non-sparse activation functions like GLU variants (Shazeer, 2020) since the resulting LMs with such activations have competitive performance in downstream tasks. Because of this, it is difficult for neurons to have no contribution to the output since these functions do not explicitly have an interval that maps to zero.
>
> Our main contribution is to propose an adaptive activation that can be exactly zero at times with little to no performance loss. We focus on the (SiLU, ReLU) pair of activations since the SiLU activation is a core ingredient of  highly competitive LLMs (Touvron, 2023).  ReLU is highly sparse but it has a performance gap with respect to SILU, e.g. there exists a tradeoff between these two activation functions.
>
> To answer the reviewers question, our methods are generalisable and can be applied to other activation functions in the following way:
>
> i) The StochA method is generalisable since one can take any non-sparse activation (which takes non-zero values on the negative side) and convert it into a stochastic activation function that takes the form of either of the activations on the positive side, and then, on the negative side either take the non-zero value of the non-sparse variant with probability p or is set to zero with probability 1-p.
>
> ii) the Swi+FT method is also generalisable since we can take an LM with a non-sparse activation like tanh, train it for 90% of the steps and then finetune it with ReLU (which is the example the reviewer mentioned).
>
> To show how both our methods would work with the (Tanh,ReLU) case, we pretrain a 1.5B LM with tanh as the non-sparse activation and relu as the sparse one for i) and ii), and compare the losses to the corresponding baselines using only the tanh or only the relu activations. We denote the activation for i) with Tanh+FT with parameter alpha and for ii) with [T|R]-T+ with parameter p, respectively.
>
> What we observe is that  **there is barely any gap between the losses of the non-sparse (tanh, bolded) and the sparse (relu, in red) activations, so there is no tradeoff between sparsity and performance, in contrast to the (SiLU,ReLU) case**. Furthermore, both StochA and Tanh+FT training losses fall above the two baseline losses:
>
> |(Tanh-ReLU-table)| activation function   |   train loss |
> |:-----------------|:------------------|-------------:|
> | | Tanh              |      **2.133**  |
> | | ReLU              |      $\color{red}{2.140}$ |
> | | Tanh+FT,alpha=0.1 |      2.193 |
> | | [T\|R]-T+, p=0.5   |      2.314 |
> | | [T\|R]-T+, p=0.3   |      2.368 |
>
> The reported train losses were averaged over the last 500 steps. Nevertheless, we are including this extra experiment in the Appendix i) to illustrate that both the StochA and Swi+FT are generalizable and ii) To contrast it to the (SiLU, ReLU) pair and emphasize why the  (SiLU, ReLU) is noteworthy. We appreciate the reviewer's suggestion.
>
> References
> - N. Shazeer. Glu variants improve transformer. arXiv preprint arXiv:2002.05202, 2020
> - Touvron H. et al. LLaMA: Open and Efficient Foundation Language Models. arXiv:2302.13971, 2023
>
> [continues in next comment]

---

> ### Author Response · Authors · 2025-11-25
>
> *Can you comment on when/how your method is preferred over post-training sparsity?*
>
> We highlight that method can be leveraged starting with a pretrained model.  We ran an experiment where we do continuous pretraining (CPT) on a 1.5B LM pretrained with the SiLU activation. During CPT we use i) Swi+FT, ii) StochA with p=0.5 or iii) StochA+Swi+FT with p=0.5 and alpha=0.1. We use 45629 steps, which is the same number of steps used for pretraining. We compare the losses against the baseline CPT model with SiLU.
>
>  In contrast to the pretrained models, the best strategy is to use Swi+FT, i.e., simply fine-tune with RELU the model initially trained with SILU to obtain the best trade-off between sparsity and loss.
>
>
> |                  | pretrain   | CPT       | inference   | StochA: p    |   Sparsity (%) | train loss |
> |:------------------------------|:-----------|:---------|:---------|:---------|---------------:|----------------------:|
>  |   baseline                  | SiLU       | SiLU      | SiLU              |                 |    0.0002 |         **2.076** |
> |   Swi+FT          | SiLU       | ReLU      | ReLU         |            |   **90.96**     |          $\color{red}{2.086}$   |
>  | StochA_0.5                | SiLU       | [S\|R]-S+    | ReLU         | 0.5          |   41.02     |2.095|
> |StochA_0.5+FT | SiLU       |  ([S\|R]-S+)+FT | ReLU          | 0.5         |   $\color{blue}{81.61}$     |      2.096 |
>
>
> We thank the reviewer for the references provided and we are incorporating the relevant ones in the related work in the new version of the paper.

---

### Official Review · Reviewer_rJBi · 2025-10-27

**Soundness:** 3
**Presentation:** 2
**Contribution:** 1
**Rating:** 4
**Confidence:** 4

**Summary:**

This paper introduces two distinct methods for training with multiple activation functions. The first method, Swi+FT, is a deterministic two-stage process: the model is first trained with SiLU, then the final portion of training is completed with ReLU. The goal is to achieve the high quality of a SiLU-trained model while retaining the sparsity and inference-speed benefits of ReLU, which the authors demonstrate on CPUs. The second method, StochA, trains using a stochastic activation that randomly interpolates between ReLU and SiLU (controlled by a probability p). This is proposed as a novel, inference-time tool to control output diversity, similar to temperature sampling. Both methods achieve performance similar to SiLU, with Swi+FT being 1.65x faster on CPUs and StochA providing activation swapability as a tool for increasing generation diversity.

**Strengths:**

- The paper successfully introduces and validates two distinct, novel methods for training with multiple activation functions.
- The primary method, Swi+FT, is well-motivated and its "best of both worlds" claim is supported by the data. The results clearly show it produces a final ReLU model with the high test-set quality of a SiLU-trained model, while retaining ReLU's sparsity for a 1.65x speedup on CPUs.
- The experiments are robust, validating these findings across two different model scales (1.5B and 3B) on various downstream tasks.
- The results show a positive scaling trend: the quality benefit of Swi+FT over the ReLU baseline increases with the larger model, suggesting its value may grow with model size.
- The second method, StochA, is also shown to be effective, achieving SiLU-level performance while providing a new, functional lever for controlling output diversity at inference time.

**Weaknesses:**

- W1. The 1.65x speedup from the Swi+FT method is demonstrated exclusively on CPUs. This limits the work's impact, as large-scale models are predominantly deployed on GPUs/TPUs where the benefits of activation sparsity are often different or less pronounced. For true "edge device" applications, this "decent" speedup is likely insufficient, as activation sparsity is only one part of a much larger optimization problem.
- W2. The diversity claim for Stocha is not convincingly supported or motivated. The diversity analysis fails to compare StochA against the most common and simple tools we already have, namely temperature and nucleus (top-p) sampling. It is unclear if StochA offers any real benefit over these established methods.
    - The evaluation also lacks qualitative examples or application to a task where controlled diversity is critical (e.g., RL exploration), making it hard to judge the usefulness of this new diversity lever.
     - Finally, the paper is missing comparisons to highly relevant prior work. For the StochA method, there is no comparison to other stochastic activation functions (like ASH).
- W3. More broadly, the paper doesn't make a strong case for its novelty in a somewhat saturated field, and it's unclear if this complex approach is truly superior to simpler, established adaptive activation functions.

**Questions:**

- The paper's motivation is weakened by the lack of clear justification for the complexity of its novel activation function compared to simpler stochastic regularization techniques. The authors do not discuss why this approach is necessary or superior to existing stochastic dropout methods (like Layer Dropout) which offer clearer, multi-faceted benefits in practice (e.g., dynamic model sizing or uncertainty estimation).
- To validate the quality claims of the $\text{Swi+FT}$ method, could the authors confirm that the total training compute (total number of iterations) was held constant or until convergence across the $\text{SiLU}$, $\text{ReLU}$, and $\text{Swi+FT}$ baselines?
- What are the benefits of $\text{StochA}$ compared to the most widely used diversity levers (e.g., temperature sampling and top-p sampling? Evaluating on a use case where controlled diversity is critical as in RL for LLMs would be helpful.
-  The practical benefits of the StochA method need stronger justification. The authors should explicitly discuss and quantify how the benefits of StochA (controlled diversity) compare with existing diversity methods such as temperature and top-p sampling.
- The Related Work section contains an editorial comment that should be removed for the final version.

---

> ### Author Response · Authors · 2025-11-25
> **Part I: reply to reviewer**
>
> We thank reviewer for their comments.
>
> *Could the authors confirm that the total training compute (total number of iterations) was held constant or until convergence across SiLU ReLU and Swi+FT  baselines?*
>
> We confirm that the total training compute is the same, more precisely, we use the same number of steps when we train the LM with each activation variant. For the 1.5B LM, we use 45,629 steps in total and for the 3B LM, 80,000 steps. For example, for the Swi+FT, we pretrain for 41,066 and switch to ReLU for the remaining 4,563 when we choose to finetune for the last 10% steps (for the 3B this would mean that we pretrain for 72000 and finetune for the last 8000 steps)
>
> *The practical benefits of the StochA method need stronger justification, explicitly discuss and quantify how the benefits of StochA (controlled diversity) compare with existing diversity methods such as temperature and top-p sampling. What are the benefits of StochA compared to the most widely used diversity levers?*
>
> Firstly, we would like to clarify that Figure 6 already includes a comparison with respect to standard temperature sampling. In Figure 6, we show the performance of a 3B LM with the StochA activation at both training and inference time for the NQ and TQA generation tasks. We compare it to a baseline 3B LM with the SiLU activation and standard temperature sampling: **we get better performance with respect to the f1 metric when we use StochA if we rank the generations by NLL.**
>
> We conducted an additional experiment to have a quantitative assessment of the diversity of generations. We report the type-token ratio metric (TTR) to complement the plots presented in Figure 6 using the same experimental setup:
>
> Experimental setup. We use the LM 3B models trained with the StochA activation [S|R]-S+ with p in {0.3, 0.5, 0.7, 0.85}. At inference time, we use the corresponding StochA activation function with the same probability used at training time. We then generate 20 answers for the Natural Questions and TQA generation tasks, compute the type token ratio of the corresponding n generated answers (samples) per datapoint and compute the average over all datapoints for a given number of samples n in {2,3,4,5,10,20}. We compare against standard temperature sampling using SiLU activation with temperature temp in {0,0.1}.
>
> Results. **We observe that the models that use the StochA activation functions have higher average TTR for a given temperature, consequently, these generate more diverse answers than the baseline with SiLU using standard temperature sampling**:
>
> - For temp=0, StochA with p=0.5 has the highest average TTR across all number of samples for the NQ and TQA tasks
> - For temp=0.1, StochA with p=0.3 has the highest average TTR across all number of samples for the for the NQ task and Stocha with p=0.5, for the TQA task
>
> For the NQ task, temp=0:
>
> | number of samples          |  SiLU |  StochA, p=0.3 | StochA, p=0.5 |   StochA, p=0.7 |   StochA, p=0.85 |
> |:-----------|------------------:|------------------------:|------------------------:|------------------------:|-------------------------:|
> | 2  |         0.493   |                0.657 |               **0.656** |                0.644 |                 0.629 |
> | 3  |         0.329  |                0.517 |                **0.509** |                0.497 |                 0.478 |
> | 4  |         0.247   |                0.438 |                 **0.421** |                0.414 |                 0.390 |
> | 5  |         0.197  |                0.386 |                 **0.365** |                0.357 |                 0.335 |
> | 10 |         0.099 |                0.259 |                 **0.228** |                0.225 |                 0.205 |
> | 20 |         0.049 |                0.176 |                 **0.139** |                0.140 |                 0.123 |
>
> For the NQ task, temp=0.1:
>
> |     number of samples      |   SiLU |  StochA, p=0.3 | StochA, p=0.5 |   StochA, p=0.7 |   StochA, p=0.85 |
> |:-----------|--------------------:|--------------------------:|--------------------------:|--------------------------:|---------------------------:|
> | 2  |            0.617 |                  **0.694** |                  0.684 |                  0.679 |                   0.671 |
> | 3  |            0.4645 |                   **0.570** |                  0.546 |                  0.544   |                   0.53 |
> | 4  |            0.380 |                   **0.495** |                  0.464 |                  0.463 |                   0.451 |
> | 5  |            0.325 |                   **0.444** |                  0.410 |                  0.407 |                   0.394 |
> | 10 |            0.200 |                   **0.315** |                  0.275 |                  0.275 |                   0.260  |
> | 20 |            0.122 |                   **0.223** |                  0.179 |                  0.184 |                   0.169 |
>
>
>
> [continued in next comment]

---

> ### Author Response · Authors · 2025-11-25
> **Part II: reply to reviewer**
>
> For the TQA task, temp=0:
>
> |          number of samples      |   SiLU |  StochA, p=0.3 | StochA, p=0.5 |   StochA, p=0.7 |   StochA, p=0.85 |
> |:-----------|------------------:|------------------------:|------------------------:|------------------------:|-------------------------:|
> | 2  |         0.498  |                0.607 |                **0.614** |                0.609 |                0.591  |
> | 3  |         0.332  |                0.450 |                **0.459** |                0.449  |                0.433  |
> | 4  |         0.249  |                0.363 |                **0.371** |                0.363  |                0.346  |
> | 5  |         0.199  |                0.307 |                **0.312** |                0.307 |                0.290  |
> | 10 |         0.100 |                0.180 |                **0.183** |                0.181 |                0.166  |
> | 20 |         0.050 |                0.104 |                **0.106** |                0.104 |                0.095 |
>
> For the TQA task, temp=0.1:
>
> |     number of samples       |   SiLU |  StochA,p=0.3 |   StochA, p=0.5 |   StochA, p=0.7 |   StochA, p=0.85 |
> |:-----------|--------------------:|--------------------------:|--------------------------:|--------------------------:|---------------------------:|
> |2  |           0.568  |                  0.627 |                  **0.634** |                  0.623  |                   0.616 |
> | 3  |           0.405  |                  0.474 |                  **0.479** |                  0.471 |                   0.458 |
> | 4  |           0.320  |                  0.389 |                  **0.392** |                  0.385 |                   0.370 |
> | 5  |           0.265  |                  0.332 |                  **0.333** |                  0.328 |                   0.314 |
> | 10 |           0.147  |                  0.199 |                  **0.202**  |                  0.198 |                   0.187 |
> | 20 |           0.082 |                  0.119 |                  **0.121** |                  0.117 |                   0.110 |
>
> *The evaluation also lacks qualitative examples.*
>
> In Appendix D, Table 7, we had already included qualitative examples of diverse generations if we use a 3B LM with the StochA activation functions at training and inference time. We added a reference to such an Appendix in the main text.
>
> *Why is this approach necessary or superior to existing stochastic dropout methods?*
>
> Using the StochA activation functions is similar to dropout but only applied it the negative side of the input after activation: with probability 1-p the negative side is equal to zero and with probability p it takes a non-zero value (given by the SiLU function on the negative side). Furthermore, since we can leverage StochA activations at inference time,  we can obtain an uncertainty quantification for each generated token if we do multiple forward passes as in Monte Carlo dropout (Gal and Ghahramani, 2016).
>
> References
> - Gal, Y. and Ghahramani, Z. (2016) "Dropout as a Bayesian Approximation: Representing Model Uncertainty in Deep Learning," ICML

---

### Official Review · Reviewer_bZvR · 2025-11-02

**Soundness:** 3
**Presentation:** 2
**Contribution:** 4
**Rating:** 4
**Confidence:** 4

**Summary:**

This paper introduces two new methods to leverage the expressivity of SiLU while, at the same time, the sparsity and computational efficiency of ReLU at inference time. The first method Swi+FT trains on SiLU, then adapts the network to ReLU for the last 5-10% of training. The second method StochA samples the activation as Bernoulli(p) during training or test time. The product is a ReLU-based model at inference time that outperforms ReLU-based training.

**Strengths:**

I enjoyed this paper for several reasons. The approach seems **novel** (I may be wrong, I'm not an expert in this field), yet **simple**. The paper is a largely empirical one: the authors did **extensive experiments** which validate their main thesis that SiLU's expressivity can be exploited during training to make better models; then ReLU's sparsity can be exploited at inference time to make them fast.

**Weaknesses:**

The reason my score is not higher is for the following reasons. In its current state, the paper is quite difficult to read and some evaluation points not justified. If both points are resolved (should be easy to fix), I will increase my score to "accept".

1. The main area where the paper can be improved is the presentation. Overall, there are many experimental results, which makes the takeaways get lost. For instance, Table 2 is extremely difficult to parse. Please distill Table 2 to the key results most salient for the paper, add **bolding** for the best performances, and move the rest to the appendix. In general, visually highlighting the takeaways for each paragraph of subsection is necessary given the density of the results.

2. Diversity of generations ablations. Why does the F1 score signal higher diversity? I would have expected to see an actual text diversity metric, e.g., type-token ratio, entropy, etc.

**Questions:**

l271 Typo: The following only applies to the StochA strategy: we evaluate the performance when if leverage the randomness at test time

---

> ### Author Response · Authors · 2025-11-25
> **Part I: Reply to reviewer's feedback**
>
> We thank the reviewer for providing actionable feedback and positive reviews for our work.
>
> *Diversity of generations ablations, actual text diversity metric missing*
>
> Thank you for the suggestion. We conducted this experiment to have a quantitative assessment of the diversity of generations. We report the type-token ratio metric (TTR) to compliment the plots presented in Figure 6 using the same experimental setup.  In Figure 6, we showed that we get better performance when we use StochA, with respect to the f1 metric, if we rank the generations  by NLL.
>
> **Experimental setup.** We use the LM-3B models trained with the StochA activation [S|R]-S+ with p in {0.3, 0.5, 0.7, 0.85}. At inference time, we use the corresponding StochA activation function with the same probability used at training time. We then generate 20 answers for the Natural Questions and TQA generation tasks, compute the type token ratio of the corresponding n generated answers (samples) per datapoint and compute the average over all datapoints for a given number of samples n in {2,3,4,5,10,20}. We compare against standard temperature sampling using SiLU activation with temperature temp in {0,0.1}.
>
> **Results.** We observe that **the models that use the StochA activation functions have higher average TTR for a given temperature, consequently, these generate more diverse answers than the baseline with SiLU using standard temperature sampling**:
>
> - For temp=0, StochA with p=0.5 has the highest average TTR across all number of samples for the NQ and TQA tasks
> - For temp=0.1, StochA with p=0.3  has the highest average TTR across all number of samples for the for the NQ task and Stocha with p=0.5, for the TQA task
>
> For the NQ task, temp=0:
>
> |(NQ-Table-1) | number of samples  |  SiLU |  StochA, p=0.3 | StochA, p=0.5 |   StochA, p=0.7 |   StochA, p=0.85 |
> |:-----------|:------|------------------:|------------------------:|------------------------:|------------------------:|-------------------------:|
> | | 2  |         0.493   |                0.657 |               **0.656** |                0.644 |                 0.629 |
> | | 3  |         0.329  |                0.517 |                **0.509** |                0.497 |                 0.478 |
> | | 4  |         0.247   |                0.438 |                 **0.421** |                0.414 |                 0.390 |
> | | 5  |         0.197  |                0.386 |                 **0.365** |                0.357 |                 0.335 |
> | | 10 |         0.099 |                0.259 |                 **0.228** |                0.225 |                 0.205 |
> | | 20 |         0.049 |                0.176 |                 **0.139** |                0.140 |                 0.123 |
>
> For the NQ task, temp=0.1:
>
> |(NQ-Table-2)  |     number of samples      |   SiLU |  StochA, p=0.3 | StochA, p=0.5 |   StochA, p=0.7 |   StochA, p=0.85 |
> |:-----------|:-----------|--------------------:|--------------------------:|--------------------------:|--------------------------:|---------------------------:|
> | | 2  |            0.617 |                  **0.694** |                  0.684 |                  0.679 |                   0.671 |
> | | 3  |            0.4645 |                   **0.570** |                  0.546 |                  0.544   |                   0.53 |
> | | 4  |            0.380 |                   **0.495** |                  0.464 |                  0.463 |                   0.451 |
> | | 5  |            0.325 |                   **0.444** |                  0.410 |                  0.407 |                   0.394 |
> | | 10 |            0.200 |                   **0.315** |                  0.275 |                  0.275 |                   0.260  |
> | | 20 |            0.122 |                   **0.223** |                  0.179 |                  0.184 |                   0.169 |
>
> For the TQA task, temp=0:
>
> |(TQA-Table-1)|  number of samples   |   SiLU |  StochA, p=0.3 | StochA, p=0.5 |   StochA, p=0.7 |   StochA, p=0.85 |
> |:-----------|:-----------|------------------:|------------------------:|------------------------:|------------------------:|-------------------------:|
> | | 2  |         0.498  |                0.607 |                **0.614** |                0.609 |                0.591  |
> | | 3  |         0.332  |                0.450 |                **0.459** |                0.449  |                0.433  |
> | | 4  |         0.249  |                0.363 |                **0.371** |                0.363  |                0.346  |
> | | 5  |         0.199  |                0.307 |                **0.312** |                0.307 |                0.290  |
> | | 10 |         0.100 |                0.180 |                **0.183** |                0.181 |                0.166  |
> | | 20 |         0.050 |                0.104 |                **0.106** |                0.104 |                0.095 |
>
> [continued in next comment]

---

> ### Author Response · Authors · 2025-11-25
> **Part II: reply to reviewer's feedback**
>
> For the TQA task, temp=0.1:
>
> |(TQA-Table-2)|     number of samples       |   SiLU |  StochA,p=0.3 |   StochA, p=0.5 |   StochA, p=0.7 |   StochA, p=0.85 |
> |:-----------|:-----------|--------------------:|--------------------------:|--------------------------:|--------------------------:|---------------------------:|
> | |2  |           0.568  |                  0.627 |                  **0.634** |                  0.623  |                   0.616 |
> | | 3  |           0.405  |                  0.474 |                  **0.479** |                  0.471 |                   0.458 |
> | | 4  |           0.320  |                  0.389 |                  **0.392** |                  0.385 |                   0.370 |
> | | 5  |           0.265  |                  0.332 |                  **0.333** |                  0.328 |                   0.314 |
> | | 10 |           0.147  |                  0.199 |                  **0.202**  |                  0.198 |                   0.187 |
> | | 20 |           0.082 |                  0.119 |                  **0.121** |                  0.117 |                   0.110 |
>
>  *The presentation can be improved, please distill Table 2  to the key results most salient for the paper, add bolding for the best performances, and move the rest to the appendix.*
>
> Thank you for the suggestion, we are working on improving the v2 version of the paper before the rebuttal period ends and plan to incorporate this feedback. Our main goal for Table 2 was to show that we mostly match the topline (SiLU) average performance. We present all the separate evaluation benchmarks as well because we want to provide a more fine grained view of the effect of using the StochA activation vs ReLU at inference time for each one.  However, we agree that this table can be simplified to make it less loaded with information and some content moved to the Appendix.

---

### Official Review · Reviewer_YY3K · 2025-11-04

**Soundness:** 3
**Presentation:** 3
**Contribution:** 2
**Rating:** 4
**Confidence:** 3

**Summary:**

To balance the sparsity and efficiency of activation functions during LLM training, this paper introduces stochastic activations that randomly select ReLU and SILU. Specifically, in experiments, the first method, Swi+FT, use SILU at 10% of the training steps and ReLU for the rest steps; the second method, STOCHA, randomly selects ReLU and SILU during training and uses ReLU for inference. Experimental results show the comparable performance and reduced inference times of the proposed stochastic activations.

**Strengths:**

1. The paper is well-written and easy to follow.
2. The proposed stochastic activation is simple while addressing the sparsity and computational efficiency of ReLU.
3. The experimental results show comparable performance of stochastic activations and reduced inference time.

**Weaknesses:**

1. This paper only proposes to switch activation functions between ReLU and SILU for LLM pretraining. An in-depth analysis should be conducted: why switching the activation function during pretraining and randomly selct activation functions both retain comparable performance? How models behave differently using a single or combination of these activation functions? Can we have some theoretical explanation of stochastic activations?
2. The experiments are conducted on pretraining small-scale transformers (1.5B and 3B), which may not generalize to large-scale models. Why not apply this method to post-training?
3. As shown in Figure 5, the performance of stochastic activations only outperform SILU on specific range of $\alpha$ in 1.5B model and always underperforms SILU in 3B model. It seems the performance is fragile because it depends on dedicated selection of hyperparameter and behaves differently on different models.

**Questions:**

See Questions mentioned in the point 1 and 2 in Weaknesses.

---

> ### Author Response · Authors · 2025-11-25
> **Part I: reply to reviewer**
>
> We thank the reviewer for their suggestions. We first rectify a couple of points in their summary: the Swi+FT case uses the SiLU activation during the first 90% or 95% of the training steps and then switches to ReLU for the last 10% or 5% steps (this was stated the other way around). Furthermore, when we use the StochA activations, there are two options at inference time: use ReLU to have higher sparsity or use StochA if we are interested in reducing the train-test gap, e.g. have comparable performance to SiLU (topline).
>
> *It seems the performance is fragile because it depends on dedicated selection of hyperparameter and behaves differently on different models.*
>
> The goal of our contribution is to match the performance of SiLU (topline) while providing sparsity but not to outperform SiLU. The fact that in Figure 5, SiLU is outperformed for some hyperparameters in the LM 1.5B case was actually a surprising result.
>
> We disagree that the method is very sensitive to the choice of  hyper-parameters: there is a robust gap between p=0 (no stochastic activation) and p > 0, with a relatively flat behavior for alpha in 5%, 10% and 20%, irrespective of the model size.
>
>  Furthermore, we only leverage two hyper-parameters (p and alpha) and our extensive experiments show that they are relatively robust, e.g.we provide many tests and ablations that show consistent behavior of our conclusions across tasks, model sizes and variations, and we had many preliminary results that had also concurred with our findings.
>
> Lastly, in most deep learning techniques the hyper-parameters typically depend on the model size, see Kaplan et al. (2020)*. Therefore, it is expected for any technique that the best performance would depend on "a dedicated selection of hyperparameters".
>
> *How models behave differently using a single or combination of these activation functions?*
>
> We tried different combinations of activations, and overall, the conclusion is that it is best to use the same activation during pretraining, with some probability, as the one used at test time. The best pair in terms of usage is (SILU, RELU), since SILU offers a better optimization and RELU provides the sparsity property that we want to exploit. Other attempts at improving the pre-training phase with something different from SiLU failed.
>
> *Can we have some theoretical explanation of stochastic activations?*
>
> We don't know how it would be possible to provide an 'theoretical explanation' beyond what we already have explained intuitively and empirically in the paper.
>
> *Why not apply this method to post-training?*
>
> We thank the reviewer for this suggestion. We ran an experiment where we do continuous pretraining (CPT) on a 1.5B LM pretrained with the SiLU activation. During CPT we use i) Swi+FT, ii) StochA p=0.5 or iii) StochA+Swi+FT with p=0.5 and alpha=0.1. We use 45629 steps, which is the same number of steps used for pretraining. We compare the losses against the baseline CPT model with SiLU.
>
>  In contrast to the pretrained models, the best strategy is to use Swi+FT, i.e., simply fine-tune with RELU the model initially trained with SILU to obtain the best trade-off between sparsity and loss.
>
> |   (CPT-Table)     | pretrain   | CPT       | inference   |  StochA: p   |   Sparsity (%) | train loss |
> |:---------------|:-----------|:------------|:---------|:------------|---------------:|----------------------:|
>  | baseline                         | SiLU       | SiLU      | SiLU         |              |    0.0002 |         **2.076** |
>  |           Swi+FT                | SiLU       | ReLU      | ReLU              |             |   **90.96**     |          $\color{red}{ 2.086}$   |
>  | StochA_0.5                | SiLU       | [S\|R]-S+    | ReLU           | 0.5        |   41.02     | 2.095|
> |StochA_0.5+FT | SiLU       |  ([S\|R]-S+)+FT | ReLU           | 0.5              |   $\color{blue}{81.61}$     |      2.096 |
>
>
> References:
> - Kaplan, Jared et al. “Scaling Laws for Neural Language Models.” ArXiv abs/2001.08361 (2020): n. pag.
>
> [continues in next comment]

---

> ### Author Response · Authors · 2025-11-25
> **Part II: reply to reviewer**
>
> For completeness, we include here the sparsity ratios for the 1.5B pretrained models, (the corresponding train and val losses can be consulted in Table 1 in the paper):
>
> |  (Pretrain-sparsity-Table)                  | train       | inference   | Swi+FT   | StochA: p   | Swi+FT: alpha   |   Sparsity (%) |
> |:------------------------------|:------------|:---------|:---------|:------------|:---------|---------------:|
>  |      baselines                      | SiLU      | SiLU        | no       |             |                 |    0.0002 |
>  |                              | ReLU      | ReLU        | no       |             |                 |   **94.8**     |
>  |                      Swi+FT        | SiLU      | ReLU        | yes      |             |        0.05         |   79.9   |
>  | StochA_0.3  +FT                 | [S\|R]-S+    | ReLU        | yes       | 0.3        |   0.05              |   $\color{blue}{88.5}$      |
>  | StochA_0.5  +FT                 | [S\|R]-S+    | ReLU        | yes    | 0.5         |   0.05           | 86.5    |
>  | StochA_0.7  +FT                 | [S\|R]-S+    | ReLU        | yes      | 0.7       |   0.05          |   84.6     |
>
>
>
> *Why switching the activation function during pretraining and randomly selected activation functions both retain comparable performance?*
>
>  Our main contribution is the empirical demonstration of our findings, supported by extensive experiments which include numerous tests and ablations, which consistently show that our conclusions are robust across different tasks, model sizes, and variations.
>
> Furthermore, we can compute the empirical cumulative distribution function for the inputs to the activation functions when training the LM 1.5B model with: SiLU, ReLU, Swi+FT and StochA+FT for different values of p. We observe that there is a sudden jump in the CDF belonging to the ReLU activation, which indicates that there are many activations that are exactly zero on input of ReLU.
>
> We then investigated the rows of the linear layer W1 and it turns out that a large fraction of these are near 0 (i.e. their norm is less that 1/1000 the average norm of the matrix). This means that these rows are unused: they are dead neurons.
> In contrast, if we examine the linear layer W1 of our Swi+FT method, which uses SiLU first and then fine tunes with ReLU, these dead neurons disappear. This explains why our approach can obtain significantly better results and still leveraging ReLU for the last part of training as opposed to the vanilla case where we pretrain with ReLU from scratch.
>
> We are including both the CDF and the dead vs non-dead neurons in W1 layer plots in the new version of our paper which we will submit before the rebuttal period ends.

---

### Author Response · Authors · 2025-11-26
**Call to action: please consider our replies**

We thank all the reviewers for their time and feedback, which helped to improve our paper and elucidate the main contributions. The reviewers highlight the following strong points:

*i) We successfully introduce and validate two distinct, novel methods for training with multiple activation functions.*

*ii) We conducted extensive experiments to validate our claims.*

*iii)The experiments are robust, validating these findings across two different model scales on various downstream tasks.*

*iv) StochA, achieves SiLU-level performance while providing a new, functional lever for controlling output diversity at inference time.*

*v)Training with stochastic activations is shown to have performance on par with SiLU activations, resolving a major weakness of ReLUs.*


 We have addressed the reviewers main feedback by conducting additional experiments, these are now posted as point to point replies to each individual reviewer. We have also incorporated these results into the manuscript together with the suggested edits and paper improvement feedback. We have now uploaded the new version of the paper.

Summary of new experiments and results:

- **Quantitative metric for  diversity of generations when using StochA.** We observe that the models that use the StochA activation functions have a higher average type-token ratio (TTR) than the SiLU baseline for a given temperature, consequently, these generate more diverse answers than the baseline with SiLU using standard temperature sampling (see NQ-Table-1, NQ-Table-2,  TQA-Table-1 and  TQA-Table-2 in reply to reviewer bZvR, which correspond to Figure 7 (b) in the paper).

- **Continuous pretraining 1.5B LMs with StochA and Swi+FT.** Starting from a pre-trained 1.5B model with SiLU, we can apply either Swi+FT or StochA in a continual pre-training approach, i.e., fine-tuning with a large number of tokens. In that case, our new experiments show that the best strategy is to use Swi+FT, i.e., simply fine-tune with RELU the model initially trained with SILU to obtain the best trade-off between sparsity and loss. (see CPT-Table in reply to reviewer YY3K, which corresponds to Table 6 in paper).

- **Show how to generalize our methods with a different pair of activations (Tanh, ReLU).** We observe that there is barely any gap between the training losses of the non-sparse (Tanh) and the sparse (ReLU) activations, so there is no tradeoff between
sparsity and performance, in contrast to the (SiLU,ReLU) case. (seeTanh-ReLU-table  in reply to q6x3, which corresponds to Table 7 in paper)

- **Added sparsity ratios for the pretrained 1.5B LMs to further back up our claims.** In the Swi+FT case, fine-tuning with RELU brings back a lot of sparsity (80%), yet the model is not competitive. The combination of StochA+Swi+FT leads to models offering a high sparsity degree by using RELU at test time and simultaneously achieving competitive performance (see Tables 1 and 2 in the paper, and the Pretrain-sparsity-Table in reply to reviewer YY3K, which corresponds to Table 5 in paper)

- **Analysis for verifying the dying ReLU problem when training a LM from scratch with ReLU.** A large fraction of the rows of the linear layer W1 are near 0 (i.e. their norm is less than 1/1000 the average norm of the matrix): these are dead neurons. In contrast, if we examine the linear layer W1 of the Swi+FT method these dead neurons disappear. This offers an additional explanation why our approach can obtain significantly better results and still leverage ReLU for the last part of training as opposed to pretraining with ReLU from scratch. (Plot added in Figure 7 (right) in paper)

- **Analysis of the empirical cumulative distribution functions for the inputs to the activation.** For x<0, the probability P(X ≤x) is lower for SILU than for both Swi+FT and when we combine StochA with Swi+FT for different values of p. Hence, StochA with Swi+FT enables us to assign more mass to negative values of x and control  it with the hyperarameter p. (Plot added in Figure 7 (left) in paper)

---

### Meta-Review · Area_Chair_Uc3j · 2026-01-06

**Summary:**

The reviewers generally see this as a well-executed empirical contribution, with praise for the method's simplicity, extensive experiments, and the practical appeal of sparse, fast ReLU models. Significant concerns remain about the practical impact on GPU/TPU inference, and positioning of the StochA for promoting sample diversity compared to alternative methods. The rebuttal provides additional analysis and experiments (including shoring up the diversity expts) as well as commitment to improve the paper's presentation. Response to questions about GPU/TPU inference is notably absent.

**Reviewer Concerns:**

The rebuttal addresses several issues: compute parity across baselines, diversity metrics (TTR) with comparisons to temperature sampling, ablations/analyses on sparsity and dead neurons. The question of performance on accelerators (GPU/TPU) remains unaddressed. While parts of the paper's contributions related to sparsity and diversity would still hold up even if accelerator performance is unimpressive on GPU/TPU, the lack of empirical results or any explicit discussion of accelerator performance is a major missing piece of the analysis, despite multiple reviewer questions on this point.

**Reviewer Scores:**

YY3K -  might move a bit; concerns partly addressed but concerns about fragility could remain

bZvR – likely mild upward move

rJBi – GPU questions unaddressed

q6x3 – GPU questions unaddressed

---

### Decision · Program_Chairs · 2026-01-26

Reject